# Progress of Research on Phytohormone Interaction in Germination of Direct-Seeded Rice under Submergence

Hui Wu [1,2,†] 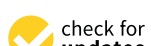, Hua Huang [1,†], Xuhui Wang [3,†], Haifang Dai [1], Yikai Zhang [2], Yaliang Wang [2], Yuping Zhang [2], Defeng Zhu [2], Huizhe Chen [2] and Jing Xiang [2,*]

[1]   Faculty of Agriculture, Forest and Food Engineering, Yibin University, Yibin 644000, China
[2]   China National Rice Research Institute/State Key Laboratory of Rice Biology, Hangzhou 310006, China
[3]   Jiaojiang District Agricultural Technique Extension Station, Taizhou 318000, China
*   Correspondence: xiangjing@caas.cn; Tel.: +86-189-5807-1661
†   These authors contributed equally to this work.

**Abstract:** Due to rainfall, closed weeding of paddy fields and other reasons, submergence stress often occurs during the germination and emergence stages of direct-seeded rice (*Oryza sativa* L.), which leads to intensified anaerobic respiration, accelerated consumption of stored nutrients, difficulty in germination of rice seeds, uneven emergence of seedlings and varying yields. Recent advances in the understanding of phytohormone interaction and the regulation of signaling pathways in crops have increased the feasibility of modulating responses to phytohormones in crop plants to enhance adaptation to environmental changes. In this review, we summarize recent advances and progress in the understanding of the regulation of phytohormone signaling pathways and their interactions with diverse internal and external signaling cues under submergence. We also discuss how these physiological modulations of phytohormones and their abundant signaling crosstalk can be applied to enhance the submergence tolerance of direct-seeded rice during germination through the manipulation of seedling morphogenesis and the fine-tuning of stress responses. Finally, we discuss how complex phytohormone signaling pathways could regulate the metabolism of stored nutrients, anaerobic respiration and energy supply in submerged direct-seeded rice seeds, thereby improving their submergence tolerance. This review hopes to provide a basis for studies of the tolerance mechanisms of submerged direct-seeded rice and the promotion of the simplified direct-seeded rice cultivation model.

**Keywords:** direct-seeded rice; germination; phytohormones; submergence

## 1. Introduction

Rice (*Oryza sativa* L.) is the main food crop in China, where about 60% of the population relies on rice as their main calorie source. The cultivated area of rice in China accounted for about 18.5% of the global rice cultivation area [1] and reached 29.92 million hm² in 2021 [2]. However, in recent years, with the development of China's rural economy and market industry adjustments, the migration of rural young adults to cities has reduced the size of the rural labor force, which currently comprises about 19.1% of China's working-age population [3]. Therefore, due to the time-consuming and labor-intensive nature of the traditional rice seedling transplanting production model, this type of production is no longer suitable as the foundation of China's agricultural development.

Direct-seeded rice cultivation is a simplified cultivation practice that eliminates the need to raise and transplant seedlings, leading to significant advantages such as reductions in cost and manual labor, less area required for seedling fields and high efficiency. In addition, the direct-seeded rice cultivation model is conducive to mechanized operation [4,5], and it has become the first choice for farmers who lack the labor force required for traditional rice agriculture [6].

Nevertheless, several obstacles must be overcome to facilitate wider adoption of direct-seeded rice cultivation, including difficulty obtaining fully grown seedlings with optimal characteristics, severe weed damage, a shortened growth period, insufficient utilization of temperature and light resources, lodging with shallow root distribution and poor yield stability [7–9]. Among these issues, poor seedling quality, a low production rate of whole seedlings and the difficulty of controlling weeds are thought to be the primary factors leading to low and unstable direct-seeded rice yield.

Reducing the use amount and frequency of chemical herbicides for weed control is an inevitable step toward green and efficient rice production. In the wet direct-seeding cultivation mode commonly used in China, the construction of a field water layer during the emergence stage of direct-seeded rice closes off the field and prevents weed growth, but this practice often leads to the long-term submergence of rice seeds, which can lead to submergence stress. In addition, paddy fields must be carefully leveled for wet direct-seeded rice cultivation because the height difference across the paddy must not exceed 3 cm [7], but this low level of variation is often difficult to achieve, resulting in different depths of field water. At the same time, the seedling period of direct-seeded rice often corresponds with periods of frequent rainfall during the spring season. Unfortunately, long-term submergence of rice seeds often leads to extremely high rates of rotten seeds and dead seedlings, resulting in uneven emergence, deterioration of population structure and high production risk [10].

In general, the suitability of the germination environment is the primary factor that determines whether direct-seeded rice seeds grow into seedlings. Varieties suitable for direct seeding must possess a high level of resistance to hypoxia stress [11,12], allowing seeds to maintain their germination potential. Phytohormones such as gibberellin, ethylene and abscisic acid participate in and regulate the various life activities of rice seeds under submergence through their signaling pathways and the interactions between them, and thus, these phytohormones influence the submergence tolerance of direct-seeded rice during germination [13–15].

In this review, we discuss the mechanisms through which phytohormones interact and regulate the submergence tolerance of rice seeds during germination, strategies for optimizing rice seedling morphogenesis and the mechanisms through which storage materials influence germination, anaerobic respiration and energy metabolism, with the goal of providing a foundation for the study of the tolerance mechanisms of submerged direct-seeded rice and the promotion of this simplified model.

## 2. Phytohormone Interactions Regulate Seed Germination under Submergence Conditions

Rice is the only cereal crop that can germinate in an anaerobic environment. Seed germination begins as the dry seeds absorb water and swell and ends when the radicle takes shape and breaks through the seed coat. The plant seed germination process is generally divided into three stages: an initial rapid water absorption stage, a middle stable water absorption stage and a final rapid water absorption stage [16,17]. Phytohormones play an important role in the germination of rice seeds. In addition to abscisic acid (ABA) and gibberellin (GA), other hormones such as auxin (AUX), melatonin (MT) and brassinosteroid (BR) are also involved in the ABA/GA regulatory pathways and rice seed germination [18].

### 2.1. Antagonistic Actions of GA and ABA Regulate Germination and Growth of Rice Seeds

GA and ABA are considered to be very critical phytohormones in crop seed germination and stress responses. GA can wake seeds from dormancy and promote seed germination, and the GA content increases significantly in the internode tissue of submerged rice seedlings, which is considered to promote the elongation and growth of internodes and blades under submergence stress [19,20]. The signal transduction process of GA follows a negative feedback regulation loop, as shown in Figure 1. First, the conformation of GA changes after binding with the receptor protein GA-insensitive dwarf protein (GID1), and it subsequently forms a trimer by binding with various DELLA proteins. Subsequently, the

GA-GID1-DELLA trimer binds to the SCF complex. The F-box of the SCF complex combines with ubiquitin to ubiquitinate the DELLA protein, after which the ubiquitinated DELLA protein can be degraded by the 26S protease, which relieves the inhibitory effect on plant growth and realizes the regulatory effect of GA on rice germination and growth [21–23].

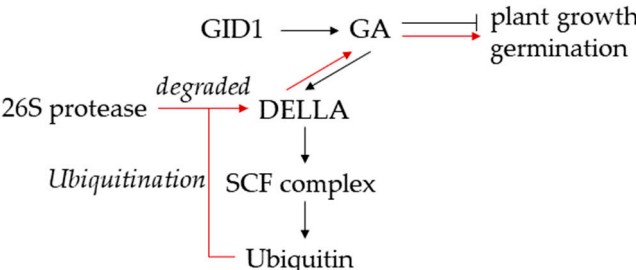

**Figure 1.** Negative feedback regulatory pathway of GA in plant germination and growth. Arrow "→" indicates induction or promotion, while bar "⊣" indicates inhibition. Black arrows indicate the binding sequence of GID1, GA, DELLA and the SCF complex, and the promoting effect of GA on seed germination and plant growth is inhibited. The red arrows indicate that after the ubiquitinated DELLA protein is degraded by the 26S protease, GA is released and plays its role in promoting germination and plant growth.

Unlike GA, which inhibits the activity of the DELLA protein and thus mediates the downstream transcriptional regulation of genes, in the presence of ABA, PYR1/PYL4 binds to PP2C and activates SnRK2s, leading to the phosphorylation of bZIP transcript factors and activation of the expression of ABA-responsive genes, including ABT (ABA Signaling Terminator). ABT interacts with both PYR1/PYL4 and PP2C proteins, such as ABI1/ABI2, and disturbs the interaction between PYR1/PYL4 and ABI1/ABI2 to decrease the inhibitory influence of PYR1 on the phosphatase activity of ABI1/ABI2. As a result, ABI1/ABI2 inhibits SnRK2s autophosphorylation, which switches off ABA signaling, leading to downstream gene transcriptional regulation [24], as shown in Figure 2.

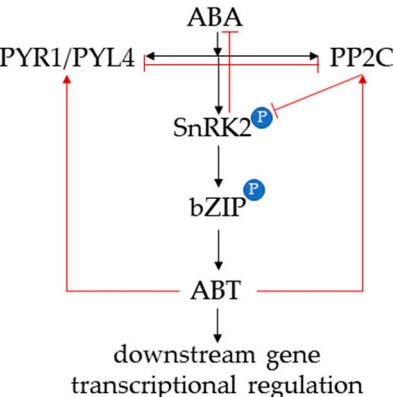

**Figure 2.** ABA signal regulatory pathway mediated by ABT. Arrow "→" indicates induction or promotion, while bar "⊣" indicates inhibition. In the presence of ABA, PYR1/PYL4 binds to PP2C and activates SnRK2s, leading to phosphorylation of bZIP transcript factors and activation of the expression of ABA-responsive genes, including ABT (ABA Signaling Terminator). ABT interacts with both PYR1/PYL4 and PP2C proteins and disturbs the interaction between PYR1/PYL4 and PP2C proteins to decrease the inhibitory influence of PYR1 on the phosphatase activity of PP2C proteins. As a result, PP2C proteins inhibits SnRK2s autophosphorylation, which switches off ABA signaling, leading to downstream gene transcriptional regulation.

It is generally believed that ABA acts as an antagonist of GA action that could inhibit shoot elongation promoted by rice seed submergence. Submergence induces the accumulation of *GA3ox1* mRNA, which is responsible for the synthesis of bioactive GAs, but

application of ABA suppresses *GA3ox1* mRNA expression in petioles of *R. palustris* [25]. There is also a direct interaction pathway between ABA and GA in the process of rice germination. The AP2 (APETALA2) transcription factor mediates the antagonism of ABA and GA. Overexpression of *OSAP2-39* upregulates the expression of *OsNCED-I*, a key enzyme in ABA biosynthesis, while the expression of the GA-inactivated gene *EUI* is also upregulated. GA plays an important role in maintaining the balance of ABA and GA in plants by regulating the activity of DELLA proteins and promoting the expression of the ubiquitin E3 ligase gene *XERICO* during abiotic stress [26,27]. In contrast, the *PLA3* gene, which encodes a glutamate carboxypeptidase with activity that is synergistic with GA, regulates seed germination by promoting GA synthesis and positively regulating GA signaling pathways [28].

### 2.2. BR Affects GA/ABA during Germination of Rice Seeds

During seed germination, the GA/ABA ratio in seeds increases via a process regulated by *OsBLR1*, a negative regulator of the BR signaling pathway. It was found that the endogenous ABA content, $GA_3$ content and $GA_3$/ABA ratio of OsBLR1-OE lines (*OsBLR1* overexpression lines in the Nip background) were 81.79%, 119.84% and 144.00% of those of germinating Nip seeds, respectively [29]. In addition, it was shown that BR inhibits ABA synthesis/transport pathway-related gene expression (*OsNCED3*, *OsSAPK3* and *OsDG1*) and negative feedback regulation of GA activity through the *OsBLR1* gene, and BR was found to enhance GA synthesis pathway-related gene expression (*OsGA20ox1* and *OsGA20ox3*) [29] (Figure 3). Interestingly, both exogenous ABA and $GA_3$ treatments increased the transcription level of *OsBLR1* in rice seeds, which promotes seed germination. ABA also inhibits BR signaling by activating the *OsREM4.1* gene in rice, and BRs inactivate *OsREM4.1* [29–31].

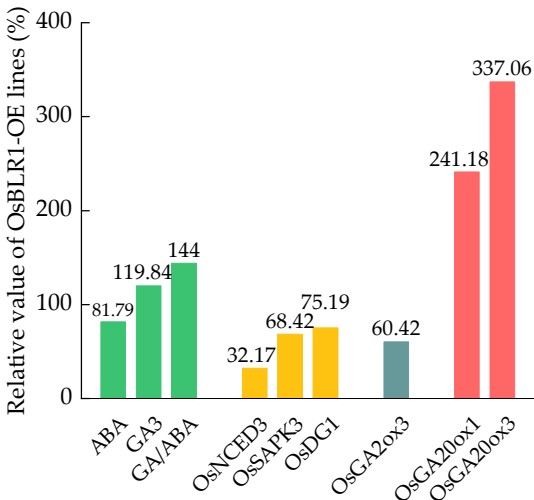

**Figure 3.** Relative values of $GA_3$, ABA and GA/ABA and gene expression levels in OsBLR1-OE lines.

### 2.3. Effects of AUX on the ABA–BR Interaction Regulation Pathway

When the ABA–BR interaction regulation pathway responds to external environmental stress, BR primarily mediates the initial stress resistance response, but the prolonged response is regulated by ABA, which is triggered by BR [32]. The transcriptional level of *DWF4*, which promotes BR biosynthesis, is also regulated by auxin (AUX). The transcriptional level of *DWF4* mRNA was shown to increase significantly after 1 h of exogenous AUX treatment, and it peaked at about four times the starting level after 12 h [33]. ABA inhibits *DWF4* expression by regulating the activity of *BIN2*, a negative regulator of BR signaling that promotes the expression of auxin-related genes by inactivating the *ARF2* auxin family transcriptional repressor, whereas enhanced BR synthesis and signaling inhibit *BIN2* activity [32]. In the presence of BR, the activity of the negative regulator *BIN2* is inhibited, thereby triggering the synergistic effect between BR and AUX [34,35].

In addition, exogenous auxin treatment of soybean seeds can significantly inhibit the expression levels of *GmGA3*, *GmGA3ox2*, *GmGA3ox1* and *GmKAO* and upregulate the transcription levels of genes such as *GmABI4*, *GmABI5* and *GmRD29-A*, thus reducing the level of active GA, increasing the level of active ABA and reducing the ratio of GA/ABA [36]. During the germination of rice seeds, the transcription of the indole acetic acid amino acid synthase gene *GH3* is also more active, which reduces the auxin signal transduction intensity during germination [37,38], thereby inhibiting the germination of rice seeds, likely by stimulating ABA signal transduction and promoting the loosening and expansion of cell walls [39,40]. In addition, there is also evidence that the AUX-responsive factor *ARF7* plays an important role in regulating the content of endogenous hormones in plants. *ARF7* may reduce the AUX/GA content and signal transduction specifically by inhibiting the expression of *GA20ox*, the AUX signal activator *ARF9* and the GA receptor gene *GID1* while promoting the expression of GA and AUX metabolic enzyme genes [41]. ABA can also promote the biosynthesis of jasmonic acid (JA) and synergistically inhibit the germination of rice seeds [42]. The interactions and regulatory signaling pathways through which GA, ABA, BR, AUX and JA influence seed germination are shown in Figure 4.

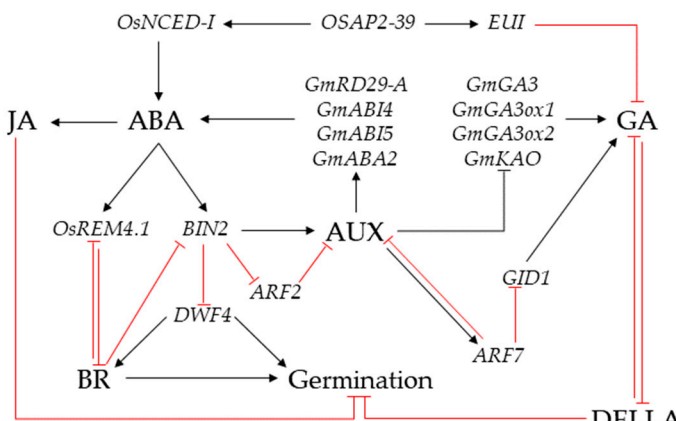

**Figure 4.** The interactions and regulatory signaling pathways through which the phytohormones GA, ABA, BR, AUX and JA influence seed germination. Arrow "→" indicates induction or promotion, while red bar "⊣" indicates inhibition. Overexpression of *OSAP2-39* upregulates the expression of *OsNCED-I*, a key enzyme in ABA biosynthesis, while the expression of the GA-inactivated gene *EUI* is also upregulated, resulting in a decrease in GA activity. ABA inhibits *DWF4* expression by regulating the activity of *BIN2*, a negative regulator of BR signaling that promotes the expression of auxin-related genes by inactivating the *ARF2* auxin family transcriptional repressor, whereas enhanced BR synthesis and signaling inhibit *BIN2* activity. ABA also inhibits BR signaling by activating the *OsREM4.1* gene in rice, and BR inactivates *OsREM4.1*. In the presence of BR, the activity of the negative regulator *BIN2* is inhibited, thereby triggering the synergistic effect between BR and AUX. Auxin could significantly inhibit the expression levels of *GmGA3*, *GmGA3ox2*, *GmGA3ox1* and *GmKAO* and upregulate the transcription levels of *GmABI4*, *GmABI5* and *GmRD29-A*, thus reducing the level of active GA and increasing the level of active ABA. The AUX-responsive factor *ARF7* may inhibit the GA receptor gene *GID1* while promoting the expression of GA and AUX metabolic enzyme genes. ABA can also promote the biosynthesis of jasmonic acid (JA) and synergistically inhibit the germination of rice seeds.

## 3. Phytohormone Crosstalk Influences the Morphological Development of Submerged Rice Seeds

### 3.1. Submergence Tolerance Strategy of Rice Seeds and Seedlings

Rice coleoptiles are one of the very few plant tissues that can grow under submergence conditions. When rice seeds germinate in hypoxic or even completely anaerobic environments, the coleoptiles of submergence-tolerant varieties are longer than those of submergence-sensitive varieties, and the growth of roots and leaves is inhibited, while

the coleoptiles are rapidly elongated. This strategy allows seedlings to obtain oxygen as early as possible, thus providing the necessary physiological and metabolic conditions for survival [43,44]. However, Setter et al. found that the excessive and rapid elongation of coleoptiles did not improve the survival rate of submerged seedlings; their study did not identify a correlation between coleoptile length and survival. In order to maintain the growth rate of coleoptiles, some varieties show excessive consumption of storage matter, which in turn reduces seedling viability [45]. During the germination stage of rice seeds, expression of the protein kinase gene *CIPK15* is upregulated through the sugar signaling pathway to generate more energy and promote the rapid growth of coleoptiles to cope with submergence stress, and this process is known as the "escape strategy" [46].

In addition, there is a submergence tolerance gene, *Sub1*, on rice chromosome 9, which plays a role distinct from that of *CIPK15*. The submergence response regulated by *Sub1* is known as the "quiescence strategy". By regulating the reaction mediated by ethylene and GA, *Sub1* inhibits the vegetative growth of rice seedlings during submergence, conserving energy and promoting recovery after submergence stress [47]. Some scholars have found that more than 70% of the phenotypic variation of submergence tolerance is associated with *Sub1*, suggesting that it may be the most important submergence tolerance gene in rice [48]. The *Sub1* locus contains three ethylene response factors, *Sub1A*, *Sub1B* and *Sub1C*, and *Sub1A* is only present in the *Sub1* QTL of the submergence-tolerant genotype [49].

### 3.2. Germination and Seedling Morphogenesis Are Regulated by Phytohormone Interaction

Phytohormone interactions play an important role in rice germination and the morphological development of rice seedlings. Several growth indicators of cereal crops, such as germination potential, shoot length and root/shoot ratio, are significantly affected by various exogenous phytohormones [50,51]. GA is the core phytohormone regulating rice morphogenesis under submergence stress. *OsEIL1* is a signal transcription factor regulated by ethylene, which can directly promote the transcriptional activity of the dwarf gene *SD1* and activate the GA synthesis pathway, accelerating the elongation of the mesocotyl and promoting the rapid emergence of submerged rice seeds, thus improving the germination efficiency of direct-seeded rice [52]. Ethylene induces expression of the *SK1* and *SK2* genes, both of which can trigger internode elongation in rice via GA. *Sub1A* improves the submergence tolerance of rice by inhibiting ethylene synthesis and GA transduction, thus reducing the consumption of stored carbohydrates in rice seeds [53]. In comparison with the GA/ABA ratio, the ZR/IAA ratio was found to be a suitable growth indicator for non-*Sub1* genotypes, and it was found to be more suitable than the former ratio as an indicator of the submergence characteristics of IR64-*Sub1* [54].

*Sub1A* can also differentially regulate BR synthesis-related genes. Significant evidence suggests that the BR level of the *Sub1A* genotype is higher than that of the intolerant genotype under submergence treatment, and the regulatory mechanism of BR–GA interaction varies under different stages of submergence stress. In the early stage of submergence, BR was found to induce the expression of GA catabolism genes, which reduced the level of GA in *Sub1A* genotype rice; in the later stage of submergence, BR mainly induced the synthesis of SLR1 protein to inhibit the GA response [55]. In addition to being affected by the submergence period, the regulatory effect of BR on GA is also related to its concentration. Low concentrations of BR promote GA synthesis, while high concentrations of BR reduce GA levels and plant height. In addition, BR was found to reduce the size of the root meristem and the number of meristematic cells, while it promoted root apical meristematic cell elongation but decreased the length of mature root cells. Moreover, BR is a potent regulator of plant morphological development. Studies have found that even 1 µmol/L of BR can significantly inhibit root elongation, and higher BR concentrations can reduce root length by about 70% [56–58]. The effects of phytohormone interaction on the germination and morphological development of submerged rice are shown in Figure 5.

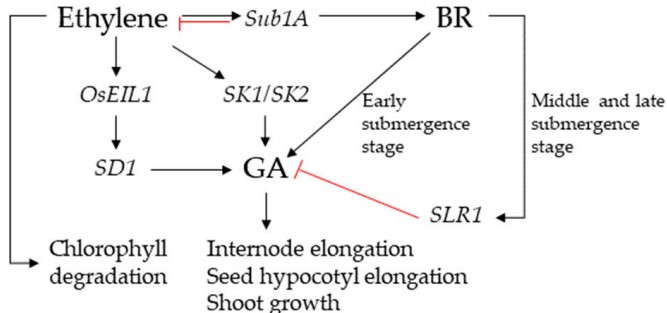

**Figure 5.** Effects of phytohormone interaction on the germination and morphological development of submerged rice. Arrow "→" indicates induction or promotion, while bar "⊣" indicates inhibition. Ethylene could induce expression of the *SK1* and *SK2* genes, regulate the signal transcription factor *OsEIL1*, directly promote the transcriptional activity of the dwarf gene *SD1* and activate the GA synthesis pathway, accelerating the degradation of chlorophyll and the elongation of the mesocotyl and promoting the rapid emergence of submerged rice seeds, thus improving the germination efficiency of direct-seeded rice. *Sub1A* inhibits ethylene synthesis and GA transduction, thus reducing the consumption of stored carbohydrates in rice seeds.

## 4. Phytohormone Interactions Regulate Storage Material Metabolism and Energy Supply

### 4.1. Phytohormones Regulate Nutrient Metabolism in Rice Seeds under Submergence

Starch stored in the endosperm is the main energy source during the germination and emergence of rice seeds. The α-amylase produced by scutellum epithelial cells or the aleurone layer is transported into the endosperm to convert starch into sugar, which is transported to growth sites to provide the carbon skeleton and energy for the formation of new tissues. However, under submergence stress, hypoxia severely impairs seed respiration and organic matter anabolism, leading to rapid consumption of storage materials, low efficiency of energy production and reduced rates of rice seed germination and seedling survival [59,60]. Recent research has shown that rice seedlings are able to recover from extended hypoxia during germination because they slow their metabolism and enter a dormant state during submergence stress [61].

Nonstructural carbohydrates (NSCs) are important because they provide the energy needed for maintenance metabolism during submergence, as well as for the regeneration and recovery of seedlings after submergence [20,62]. A critical evaluation of submergence-tolerant and submergence-intolerant rice cultivars revealed that seedlings of tolerant cultivars have 30–50% more NSCs compared to susceptible cultivars [20,63]. In this process, phytohormone interaction and regulation play key roles in the consumption and maintenance of stored substances in plant tissues. Some studies have found that submergence could promote increased ethylene levels in tissues and accelerate the degradation of starch and soluble carbohydrates. After 3 days of submergence, ethylene production by M202 continued, and the ethylene level of M202 was 1.6-fold that of M202(*Sub1*), but the total starch content of M202(*Sub1*) was only 3.27-fold that of M202 [64]. This difference may have been due to the restriction of ethylene synthesis and inhibition of the effect of GA by *Sub1A*, which would have been expected to inhibit the expression of the α-amylase and sucrose synthase genes required for sucrose and starch catabolism [64,65]. This *Sub1*-regulated pathway conserves valuable carbohydrates to prolong the survival time of rice plants in water while facilitating the recovery of plants from waterlogging [66]. In addition to ethylene, BR treatment can also lead to slower degradation of endosperm starch during rice germination. In BR-treated seedlings, the expression level of the *Wx* gene, which encodes a key enzyme of starch metabolism, was found to be downregulated by about 50%, and starch maintained better crystallinity and a more ordered structure [67].

The sugar metabolism pathway and the expression of related genes play key roles in the survival of rice seeds under submergence. For example, pyruvate metabolism plays a central role in carbohydrate metabolism, and several genes involved in this metabolic

pathway, including the pyruvate kinase gene *Os04t0677500-02*, are highly expressed under submergence stress [68]. In addition, submergence stress strongly inhibits amylase activity in rice seeds, resulting in an insufficient supply of soluble sugar and an energy deficit, which may be reversed by exogenous GAs or endogenous GAs produced by embryos. Soaking rice seeds with $GA_3$ was found to increase their amylase activity, soluble protein content, and reducing sugar content, while paclobutrazol (PP333) treatment inhibited amylase activity and reduced starch utilization, soluble protein content, and reducing sugar content, while inducing accumulation of soluble sugar [50,69].

### 4.2. Phytohormone Interactions Regulate Anaerobic Respiration and Energy Metabolism

Submergence leads to inhibition of the aerobic respiration of rice and the conversion to anaerobic respiration with a limited energy supply [70]. This process promotes glycolysis and ethanol fermentation to produce sufficient ATP to maintain the survival of seedlings. Since hypoxia-intolerant rice seedlings can also show submergence tolerance phenotypes after being fed with exogenous glucose, Huang et al. proposed that the available sugar source is an important physiological feature that determines whether a rice variety has submergence tolerance [71]. Some genes related to glycolysis are also upregulated by submergence, such as the *LOC os02g38920* gene, which encodes a glyceraldehyde phosphate dehydrogenase. Hypoxia also induces the expression of the $\alpha$-amylase gene *RAMY3D*, while gibberellin-induced amylases such as *RAMY3A* only play an important role in aerobic environments [72]. Meanwhile, when the hypoxia signal is transmitted to *CIPK15*, *CIPK15* regulates the activity of Snf1-related protein kinase 1 (*SnRK1A*), which monitors energy levels and induces the synthesis of starch hydrolases and alcohol dehydrogenases (ADH) in rice seeds through the sugar signal transmission pathway, leading to the conversion of starch into sugar and ATP, so that the seeds have sufficient carbohydrates and ATP to germinate in water [46,73]. However, under anoxic conditions in rice, far less ATP is provided by anaerobic respiration in comparison with that produced by aerobic respiration, and the carbohydrates stored in rice are therefore rapidly consumed.

Oxygen deprivation is associated with increased ethanol fermentation (catalyzed by pyruvate decarboxylase (PDC) and ADH) and lactate fermentation (catalyzed by lactate dehydrogenase (LDH)) as well as increased biosynthesis of alanine, g-aminobutyric acid (GABA), succinate and occasionally malate [74–76]. In fact, submergence dramatically elevates mRNA accumulation and enzymatic activities of pyruvate decarboxylase (PDC) and ADH and subsequently activates ethanol fermentation in rice [64,77,78]. *Adh1* is essential for sugar metabolism via glycolysis to ethanol fermentation in both the embryo and endosperm. In the endosperm, energy is presumably needed for the synthesis of amylases and sucrose synthesis, as well as for sugar transport to the embryo [76]. Notably, ethylene treatment increased the levels of transcripts encoding Pdc1, Pdc2, Pdc4, Adh1 and Adh2 in shoots of rice [64].

Phytohormones such as ethylene and substances such as $H_2S$ and $Ca^{2+}$ have the ability to regulate ethanol metabolism. Li Ying et al. found that spraying exogenous cytokinin (6-BA) after submergence can reduce the level of ABA in the roots of peanuts, increase the activities of ADH and MDH in roots and reduce the activity of LDH [79]. Under hypoxic conditions, $H_2S$ stimulates an increased endogenous $Ca^{2+}$ content in maize cells, thereby enhancing the activities of the key enzymes ADH and PDC in anaerobic fermentation, inhibiting the production of ethylene, initiating the "quiescence strategy" and thus improving submergence tolerance [80]. In *Arabidopsis thaliana* seedlings, AOA, an inhibitor of ethylene biosynthesis, significantly repressed *Adh1* transcript accumulation and enzymatic activity under hypoxia, but the *Adh1* transcript level was restored by the addition of ACC [78]. These findings suggest that ethylene contributes to the regulation of ethanolic fermentation capacity under submergence and hypoxic conditions. However, the role of ethylene in metabolism selection during germination and emergence in submerged rice seeds remains unclear.

## 5. Perspective

In consideration of the ongoing demographic changes and ecological conditions in China, the simplified direct-seeding rice cultivation model may be more suitable than traditional wet cultivation practices. Oxygen deficiency caused by seed submergence is an important factor in uneven seedling emergence and yield instability, which have hindered the expansion of direct-seeded rice technology. Through studies of interactions among plant hormones, the regulatory network underlying germination and growth in submerged rice seeds has been identified, and QTLs associated with submergence tolerance have been located, providing an empirical foundation for the breeding of rice varieties with improved submergence tolerance. A significant effort has been expended with the goal of breeding varieties suitable for submerged direct seeding from the perspective of genetic optimization. However, few studies have explored the possibility of enhancing the germination and growth of submerged rice seeds via treatment with exogenous phytohormones, despite the potential for such manipulations to allow wider and more efficient usage of existing high-quality germplasm resources. In addition, some studies have shown that the application of exogenous hormones to parent rice plants during the process of seed production can affect seed vitality under different conditions; therefore, seeds with improved tolerance for various different environmental stresses can be produced from the same parents, allowing further optimization of direct-seeded rice cultivation under a wide variety of conditions. Recent studies have shown that the regulatory network of phytohormones is extremely complex, and new regulatory factors are constantly being discovered. Many regulatory factors are regulated by upstream factors, and they may also participate in regulatory feedback loops that in turn modulate the activity of their upstream factors or promote and inhibit downstream reactions simultaneously, among other roles. Therefore, further research is warranted to dissect the complex phytohormone network underlying rice seed germination under submergence conditions with the goal of providing a foundation for improved rice cultivation and thus ensuring national and global food security in the future.

**Author Contributions:** Conceptualization, H.W. and J.X.; writing—original draft preparation, H.W and H.H.; writing—review and editing, X.W., H.D., Y.Z. (Yikai Zhang), Y.W., Y.Z. (Yuping Zhang) and H.C.; supervision, D.Z. and J.X. All authors have read and agreed to the published version of the manuscript.

**Funding:** This research was funded by the Applied Basic Research Project of Sichuan Provincial Department of Science and Technology (2020YJ0414) and the Open Project of the State Key Experiment of Rice Biology (20200403).

**Data Availability Statement:** Data sharing not applicable.

**Conflicts of Interest:** The authors declare no conflict of interest.

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
