# Peer review of "Progress of Research on Phytohormone Interaction in Germination of Direct-Seeded Rice under Submergence"

_agronomy, doi:10.3390/agronomy12102454_

Round 1

Reviewer 1 Report

The present study has the following concern before acceptance of the article for publication.

1: line 33; Keywords should be alphabetical.

2: lines 36-38; How is possible? this reference is 2015 and the information is related to 2020!

3: line 39-41; Reference?

4: lines 55-56; Reference?

5: lines 64-69; It is necessary to cite used references for a review paper. a lot of sentences are presented without citation! this information is not your interpretation because they are some facts and need to cite proper references.

6: line 101; Here readers need to study the text for understanding the Figures!! whereas "Each table and figure should stand alone, complete and informative in itself". Please Revise all the Figures and Tables, especially abbreviations.

7: lines 334-360; The perspective is very general. It is better to propose future plans in a more specialized manner according to the results of many studies that reviewed here.

Author Response

The present study has the following concern before acceptance of the article for publication.

1: line 33; Keywords should be alphabetical.

Revised:

Keywords had been revised as” Direct-seeded rice; germination; phytohormones; submergence”

2: lines 36-38; How is possible? this reference is 2015 and the information is related to 2020!

Revised:

The cultivated area of rice in China accounted for about 18.5% of the global rice cultivation area [1], and had reached 29.92 million hm2 in 2021 [2].

In this section, the latest data and relevant references have been added.

3: line 39-41; Reference?

Revised:

However, in recent years, with the development of China's rural economy and market industry adjustments, the migration of rural young adults to cities has reduced the size of the rural labor force, which currently comprises about 19.1% of China's working-age population [3].

  1. National Bureau of Statistics of China: Bulletin of the Seventh National Census (No. 5). http://www.stats.gov.cn/tjsj/tjgb/rkpcgb/qgrkpcgb/202106/t20210628_1818824.html

4: lines 55-56; Reference?

Revised: This sentence is summarized by the author and does not directly cite any reference.

5: lines 64-69; It is necessary to cite used references for a review paper. a lot of sentences are presented without citation! this information is not your interpretation because they are some facts and need to cite proper references.

Revised: The following references have been added:

  1. Septiningsih, E. M., Ignacio, J., Sendon, P.M.D., Sanchez, D. L., Ismail, A. M., Mackill, D. J. QTL mapping and confirmation for tolerance of anaerobic conditions during germination derived from the rice landrace Ma-Zhan Red. Theoretical and Applied Genetics 2013, 126, 1357-1366. [CrossRef]

6: line 111; Here readers need to study the text for understanding the Figures!! whereas "Each table and figure should stand alone, complete and informative in itself". Please Revise all the Figures and Tables, especially abbreviations.

Revised: Relevant parts of the manuscript have been revised as required. Since there are many modifications, please check the manuscript directly.

7: lines 334-360; The perspective is very general. It is better to propose future plans in a more specialized manner according to the results of many studies that reviewed here.

Revised: Relevant parts of the manuscript have been revised as required. Since there are many modifications, please check the manuscript directly.

Reviewer 2 Report

The review article by Wu et al. discussed the role of phytohormone on germination of direct-seeded rice under submergence, is well written and has scope to be publish in “Agronomy”. However, there are some issues need to be addressed before the publication of the manuscript.

Line 58, Kindly mention the others methods of non-chemical herbicides of weeds control.

Line 73. kindly mention which  “various adverse ecological environmental factors”.

Line 16, 94, Don’t use “etc” in MS, instead you can write “among others”

Figure3. Author should mentioned/cite the original paper of the data used in figure 3.

Line 138-143. this need citation.

Author Response

Line 58, Kindly mention the others methods of non-chemical herbicides of weeds control.

Revised:

Reducing the use amount and frequency of chemical herbicides for weed control is an inevitable step toward green and efficient rice production

Line 73. kindly mention which  “various adverse ecological environmental factors”.

Revised: This sentence has been deleted.

Line 16, 94, Don’t use “etc” in MS, instead you can write “among others”

Revised:Lin16, “etc.” had been revised as: and other reasons.

Lin97, “etc.” had been deleted.

Figure3. Author should mentioned/cite the original paper of the data used in figure 3.

Revised: the original paper have been added.

Line 138-143. this need citation.

Revised: Relevant references have been added.

Reviewer 3 Report

The review article is focusing on the physiological requirements of the successful cultivation of direct seeded rice through the phytohormone interactions during germination. However, direct seeding of rice is a widely utilized cultivation technique in many countries in America, Europe and Australia, the complex knowledge about the background of processes during germination under submergence stress is still significantly relevant. The article provides detailed summary of phytohormones involved in the germination process. The information is up-to-date, most of the cited literature is within 10 years. 

Only minor parts must be corrected in the reference list. References in 38, 39 and 83 are not including year of publication data. 

Author Response

Thank you very much for your careful review. The year of publication of the reference has been added.